# Diploid Wheats: Are They Less Immunogenic for Non-Celiac Wheat Sensitive Consumers?

**DOI:** 10.3390/cells11152389

**Published:** 2022-08-03

**Authors:** Vera Rotondi Aufiero, Anna Sapone, Giuseppe Mazzarella

**Affiliations:** 1Institute of Food Sciences, CNR, 83100 Avellino, Italy; vera.rotondiaufiero@isa.cnr.it; 2Center for Celiac Research and Treatment, Massachusetts General Hospital, Boston, MA 02114, USA; annasapone@yahoo.it

**Keywords:** non celiac wheat sensitivity, diploid wheat, common wheat, amylase trypsin inhibitor, FODMAP

## Abstract

Non-celiac wheat sensitivity (NCWS) is a clinical entity induced by the ingestion of gluten that leads to intestinal and/or extraintestinal symptoms, and is diagnosed when celiac disease and wheat allergy have been ruled out. In addition to gluten, other grains’ components, including amylase trypsin inhibitors (ATIs) and fermentable short-chain carbohydrates (FODMAPs), may trigger symptoms in NCWS subjects. Several studies suggest that, compared with tetraploid and hexaploid modern wheats, ancient diploid wheats species could possess a lower immunogenicity for subjects suffering from NCWS. This review aims to discuss available evidence related to the immunological features of diploid wheats compared to common wheats, and at outlining new dietary opportunities for NCWS subjects.

## 1. Introduction

Adverse reactions to food that result in gastrointestinal symptoms are common in the general population. Wheat has been found to be one of the most common factors inducing such symptoms.

In particular, some disorders are related to the ingestion of specific wheat components, such as gluten, fermentable oligo-, di-, monosaccharides, polyols (FODMAPs), and wheat amylase trypsin inhibitors (ATIs). These disorders are known as wheat-related disorders that mainly involve celiac disease (CD), wheat allergy (WA), and non-celiac gluten/wheat sensitivity [1].

CD is an immune-mediated disease triggered by the ingestion of wheat gliadin and related prolamins from other toxic cereals, such as barley and rye, that cause typical CD autoimmune enteropathy in genetically susceptible individuals [2]. WA is an IgE-mediated allergic reaction to the proteins found in wheat and other related cereals, such as barley and rye [2]. As defined by the 2015 Salerno Expert’s Criteria, the term non-celiac gluten sensitivity (NCGS) is used to describe the clinical state of individuals who develop both intestinal and extraintestinals (headache, foggy mind, chronic fatigue, joint pain, tingling or numbness of the extremities, eczema) symptoms when they consume gluten-containing foods, and feel better on a gluten-free diet (GFD), but do not have CD or a WA [3]. Subsequently, it has been recognized that some components of wheat other than gluten proteins are potentially deleterious for NCGS patients, which include ATIs and FODMAPs. The terminology “NCGS” was then changed to “Non Celiac Wheat Sensitivity” (NCWS), which would exclude other relevant cereals, such as barley and rye [4].

The limited knowledge about the pathophysiology of NCWS, and the lack of validated biomarkers, are still major limitations for clinical studies, making it difficult to differentiate NCWS from other wheat-related disorders, as well as from other clinical conditions characterized by similar symptoms (for example, irritable bowel disease). Therefore, NCWS diagnosis is still based on clinical criteria and it can be confirmed only by a double-blind placebo-controlled challenge, a practice difficult to implement in routine clinical settings [3]. Several studies suggest that NCWS is an immune-mediated disease that likely activates an innate immune response [5].

Research is actively trying to find wheat varieties with absent or low immune-reactivity to be implemented in new strategies for the treatment and prevention of subjects suffering from wheat-related disorders. Preliminary evidence supports the assumption that the diploid wheat species, *Triticum monococcum*, compared to common wheat, *Triticum aestivum*, could possess a lower immunogenic potential for subjects suffering from NCWS [5,6].

This review aims to discuss available evidence related to the immunogenic properties of diploid wheats, and outlines new dietary opportunities for patients with NCWS.

## 2. Mucosal Immune Responses in NCWS

The precise pathogenesis of NCWS is still poorly defined; nevertheless, in several reports, it has been shown to be the presence of a gut immune activation, and where innate immunity could have a role in some subjects with this condition (Table 1). In 2011, we observed that small intestine expression of toll like receptor (TLR) 2 and, to a lesser extent, TLR1, was increased in NCWS subjects, compared to CD or controls, whereas there were no differences in markers of adaptive immunity [7]. The involvement of innate immunity was confirmed by several subsequent studies, which additionally showed a higher production of cytokines that regulate innate immunity, such as interleukin (IL)-8, in mucosal biopsy specimens of NCWS patients [7,8,9,10,11,12,13,14,15,16].

Nevertheless, in an intestinal biopsy-based study, NCWS patients showed increased mucosal interferon (IFN)-γ mRNA after a 3 day gluten challenge [17]. This indicates that an adaptive immune response may also play a role in the NCWS pathogenesis. Additional compelling evidence for the role of adaptive immunity in NCWS came from other groups, showing an increased production of Tumor Necrosis Factor (TNF)-α and of IL-17, in the intestinal tissue of NCWS patients, compared to healthy individuals [18,19].

The production of antigliadin IgG antibodies, in approximately 50% of NCWS patients [7,20,21], which disappear rapidly after adhering to a GFD, together with improvement of intestinal and/or extraintestinal symptoms, support the role of the adaptive immune system in NCWS.

However, there are some reports suggesting that such an antibody response may be a consequence of impaired gut integrity and permeability [30]. In the first study on this topic, published by our group, higher levels of transcripts for Claudin (CLDN)-4, the gene associated with tight junction (TJ) function, was found in NCWS mucosa, compared to either CD or controls [7]. Subsequently, Hollon et al. [23] investigated changes in transepithelial electrical resistance (an index of intestinal permeability) in duodenal biopsy explants cultured with gliadin. The authors observed an increase in intestinal permeability in NCWS more than in CD patients and healthy subjects. An in vivo analysis of human barrier function in NCWS, was conducted by Fritscher-Ravens et al. [24], using confocal laser endomicroscopy. Here, after wheat administration, break of the TJ and infiltration of the intestinal epithelium by T-cells in patients with NCWS, were detected. In a recent study by Uhde group [22], elevated levels of multiple permeability biomarkers, such as antibodies to bacterial antigens, lipopolysaccharide-binding protein (LBP), and intestinal fatty acid-binding protein (FABP2), were found in NCWS. The authors, in agreement with previous studies, [7,20,21] also observed an increase in IgG native gliadin antibodies in NCWS, compared to the healthy control group, suggesting that such an antibody response may be due to higher small intestinal permeability. Notably, GFD leads to a normalization of these markers, demonstrating a link between diet, intestinal barrier, and systemic immune activation in NCWS patients. Additionally, gut microbiota analysis revealed a significant dysbiosis in NCWS patients, and some authors suggest that this could contribute to intestinal barrier dysfunction [31].

Other findings of gut mucosal inflammation in NCWS include an increased infiltration of eosinophils in the gastrointestinal tract [25,28]. As eosinophils are involved in both IgE-mediated and non-IgE-mediated food allergy, these studies shown that patients identified as having NCWS may have features of non-IgE-mediated food allergy. Increased levels of mast cells in the duodenum have also been reported [26], and it was suggested that such immune cells with their close vicinity to neurons, could play a role in sensory-motor dysfunction and symptom generation in patients with NCWS [27].

At present, NCWS is characterized by a normal duodenal histological picture (Marsh 0 stage), even if an increase in CD3+ IELs (Marsh I) could be detected in some patients, as reported by our own studies [7,32] and supported by others [17,20,25,28,33]. Some authors describe a peculiar lymphocytic arrangement in small intra-epithelial clusters, and a linear disposition in the deeper mucosa [25,34,35]. However, it is important to highlight that for a correct histopathological evaluation, an exact biopsies orientation, to avoid false atrophies and imprecise T lymphocyte counts, as well as the role of some infections, which can cause intraepithelial lymphocytosis, need to be taken into account.

To date, evidence to support the view that NCWS could exclusively present with normal histology or a milder enteropathy (microscopic and sub-microscopic), is lacking. We have presented an international collaborative analysis of gluten induced enteropathy, aimed to characterise a reliable diagnostic pathway for this nutrient antigen induced enteropathy, with inspiration from Salerno expert criteria [3,35].

Our findings indicate that NCWS mucosae are associated with reduced villus height, increased crypt depth, increased lymphocyte infiltration of either villi or crypts, and corresponding alterations in villus/crypt ratios, even at Marsh 0 stage. Therefore, based on this large study, we have identified mucosal alterations associated with NCWS and provided evidence that architectural distortion starts at Marsh 0 stage. Nevertheless, we were unable to verify an increase in eosinophils, as reported by other studies. Further studies are required to confirm the role of such immune cells in gluten-induced mucosal inflammation.

However, our study does provide a significant achievement that will move our understanding of NCWS forward, and brings clarity on what we consider normal intestinal mucosa, and what falls into the “microscopic enteropathy” spectrum in line with the evolutionary development study by Marsh [36]. These early and mild alterations are liable to be erroneously described as “non-specific change”, but given the right clinical and serological context, they may be helpful in supporting a diagnosis of NCWS.

Notwithstanding the findings of the studies presented above, demonstrating a gut immune activation in NCWS subjects, there are also reports that present some controversial results, leading to different interpretation. This effect is probably related to the different methodology used for NCWS diagnosis. Indeed, its diagnosis is currently based on clinical features and the double-blind placebo-controlled gluten challenge (DBPC) alone, or with a crossover arm [3] that is only rarely used in research studies due to its complexity, which could hamper the adherence of the patients. In fact, the majority of the studies performed did not employ the suggested methods of DBPC food challenge; but despite reduced diagnostic accuracy, most of the clinicians used an open gluten challenge fashion to ascertain the diagnosis of NCWS.

Therefore, further investigations are needed to identify specific biomarkers that would help clinician to diagnose NCWS in a more practical way.

## 3. Gluten Components

Wheat gluten is composed of two types of proteins called glutenins and gliadins, which in turn can be divided into high molecular and low molecular glutenins and α/β, γ and Ω gliadins [37].

Gluten composition varies between species and cultivars, presenting high contents of proline-rich polypeptide residues, which make them resistant to proteolytic degradation in the gastrointestinal tract [38]. When these proteins are consumed by genetically susceptible individuals, a cascade of immune reactions is triggered, which result in damage of the small intestinal mucosa, resulting in CD. Its pathogenesis has classically been attributed to the activation of lamina propria CD4+ Th1 cells specifically reacting to immunotoxic gluten peptides [5].

To date, a large number of peptides stimulating CD4+ T-cells have been characterized from both glutenin, α/β- and γ-gliadins [8] and, more recently, also from Ω -gliadins proteins [39]. It has been demonstrated that a 33-mer of α-gliadins that harbours six copies of three different T-cell 9-mer epitopes, is the immunodominant peptide [40]; furthermore, this peptide has a pronounced resistance to gastrointestinal enzymatic digestion that allows it to reach the intestinal immune system in an almost intact form [40]. Other studies indicate that gliadin also contains peptides able to activate an innate immune response [41,42,43]. In the early phase of CD, epithelial cells are likely destroyed via toxic gliadin peptides, such as 19-mer [29,32,33,34,35,36,37,38,39,40,41,42,43,44,45,46,47,48,49], that might activate the innate immune system, thereby upregulating interleukin IL-15 secretion [44]. Recently, it has been found that 33-mer peptide could also activate the innate immune system via TLR-2 and TLR-4 receptors, inducing the release of pro-inflammatory cytokines, such as IP-10/CXCL10 and TNF-α (Figure 1a) [45].

Therefore, since both innate and adaptive immunity are involved in CD pathogenesis, cereal suitable for a CD diet should be low in both classes of peptides. Among candidates, there are diploid wheat species, because of a reduced number of stimulatory epitopes of T-cell lines [46], and of the lack of a D-genome encoding the immunodominant 33-mer fragment [47], compared to common wheat.

*Triticum aestivum* has evolved from hybridization between the tetraploid species *Triticum turgidum* (AABB) and the diploid species *Aegilops tauschii* (DD) [47]. Due to its “simpler” genome with respect to *Triticum aestivum* and *durum*, *T. monococcum* contains a reduced number of epitopes and toxic peptides [48]. These findings may have implications for programs aiming to produce wheat species with no or low contents of gluten proteins, harmful to CD patients. Nevertheless, even if it should prove impossible to generate a wheat cultivar completely devoid of harmful proteins, a cultivar low in T-cell stimulatory sequences can possibly be tolerated by most CD patients. Moreover, the difference in gluten composition among diploid (AA), tetraploid (AABB) and hexaploid (AABBDD) wheat varieties may affect the resistant to cleavage by intestinal peptidases [49].

We have previously investigated, in in vitro models, the immunological properties of gliadin protein from two monococcum *cvs*, Monlis and Norberto-ID331, in view of their possible use in CD patients [50]. We found that partially digested gliadin proteins extracted from both monococcum lines, Monlis and Norberto-ID331, induced adaptive immune response in CD patients, whereas the innate immune response could be elicited only by gliadin from Monlis *cv* [50]. Subsequently, we have demonstrated, by proteomic analysis, that almost all immunotoxic gluten peptides from Monlis and Norberto-ID331, are in vitro degraded during digestion by gastric-duodenal and brush border membrane (GD-BBM) enzymes, whereas gluten immunogenic peptides from hexaploid *Triticum aestivum* resist intestinal digestion [51]. Moreover, *T. monococcum* gluten peptides after GD-BBM degradation, elicited a lower T-cell response compared to *Triticum aestivum* digested gluten proteins from hexaploid wheat [51]. The hypothesis of a better digestibility of diploid wheats, compared to common wheats, was confirmed by recent in vitro proteomic study [52]. Similarly, recent results showed that gliadin of *Triticum durum*, was almost unaffected by the in vitro gastrointestinal digestion, while *T. monococcum* gliadin had a marked susceptibility to digestion [53]. Clinical trials have shown that *T. monococcum* is toxic for CD patients, as judged on histological and serological criteria, but it was well tolerated by the majority of patients [54], suggesting a potential effcacy in patients suffering from other gluten-related disorders, such as NCWS. Another study underlined that in CD patients *T. monococcum* wheat elicits a reduced in vivo T-cell response compared to *Triticum aestivum*, most likely due to its higher digestibility [55].

It should be noted that these studies concern only one of the gluten-related diseases, namely CD, and that the internationally accepted guidelines currently provide that these patients should, however, avoid any type of wheat or cereal containing gluten, including *T. monococcum*. Nevertheless, results reported herein from multiple studies are encouraging findings that suggest a potential tolerability of *T. monococcum* for people suffering from NCWS.

Increased intestinal permeability in patients suffering from wheat-related disorders, could be an early event that precedes the onset of gut immune activation. In CD, it was shown that myeloid differentiation factor 88 (MyD88), a key adapter molecule in the TLR/IL-1R signaling pathways, induces release of zonulin, a mediator of gut permeability, upon non-digested gliadin binding to CXCR3 on enterocytes, as a result inducing greater epithelial permeability and subsequent paracellular gliadin passage to the gut mucosa [56]. (Figure 1a). These data support the model for the innate immune response to gliadin in the initiation of CD. Similar mechanisms may also underlie the increased intestinal permeability reported in NCWS. As we have shown that *T. monococcum* gliadin had a marked susceptibility to gastrointestinal digestion, we can hypothesize that such a mechanism, triggered by non-digested gliadin of common wheats, may not be elicited (Figure 1b). Thus, the innate immune response could be prevented.

## 4. Non-Gluten Components

Non-gluten wheat proteins comprise a mixture of components with structural, metabolic, and putative protective functions [57], such as ATIs, which comprise about 2–4% of the total wheat grain proteins, and may contribute to the defence of the plant from pests and parasites by constraining their digestive enzymes; for this reason, these proteins are highly resistant to proteases action [58]. However, they are currently among the most widely studied wheat components, because of their implication in adverse reactions to wheat consumption in humans, such as respiratory, allergy, and intestinal responses associated with CD and NCWS.

Although several studies revealed variations in the ATI content between ancient and modern wheat grains, their immunogenic potential remain elusive.

To date, ATIs have been shown to be potent activators of the innate immune system response. Junker et al. [9] found that ATIs engage TLR-4 and release of proinflammatory cytokines in myeloid cells, for example, IL-8 and IL-12, of both patients with CD and non-diseased controls, as is expected for innate immune triggers. Moreover, the authors also showed that the addition of exogenous ATIs to the organ culture of jejunal biopsies from treated CD induced an increase in IL-8 mRNA levels, compared with healthy controls. As a matter of fact, for their general TLR4 stimulatory activity, ATIs are not only a long-sought nutritional trigger of innate immunity in CD, but were suspected to have more far-reaching pathogenic roles in patients with wheat-related hypersensitivities, such as NCWS or irritable bowel syndrome (IBS) [5,59,60].

Interestingly, it was found that modern wheat contains high concentrations of ATIs, compared with ancient diploid wheat [5,61,62,63]. Therefore, considering the pro-inflammatory effect of ATIs, *T. monococcum* wheat could retain a reduced immunostimulating activity for subjects suffering from wheat related diseases.

Zevallos and co-workers found that in TLR4-responsive mouse and human cell lines, older wheat variants, such as *T. monococcum,* had lower bioactivity than modern wheat [13]. More recently, by using organ cultures of jejunal biopsies and intestinal T-cell lines from CD patients, we evaluated the immunogenic properties of ATIs obtained from two selected *T. monococcum cvs*, Monlis and Norberto-ID331, and *Triticum aestivum* spp. Sagittario *cv* after an in vitro proteolytic digestion (PC) [64]. We found that PC-digested ATIs purified from *Triticum aestivum* induced IL-8 and TNF-α secretion in organ culture of jejunal mucosa of treated CD patients, whereas the capability of ATIs from *T. monococcum* to stimulate innate immunity was meaningfully affected [14]. It has been reported that the resistance to gastrointestinal digestion is an important constrain in determining the immune stimulatory and toxicity properties of gliadin peptides [40]. Therefore, our data suggest that susceptability to enzymatic hydrolysis of ATIs from diploid *T. monococcum,* resulted in a failure to induce the innate immune response (Figure 1d). In contrast, the stability to hydrolysis by human digestive enzymes of ATIs from hexaploid wheat, affects the activation of the mucosal innate immune response (Figure 1c).

The role of ATIs as adjuvants of gluten-induced barrier dysfunction in wheat-related disorders, including CD and NCWS, has not been explored. Recently, Caminero and colleagues [64] reported that ATIs exacerbate inflammation in mice expressing HLA-DQ8, whereas in wild-type mice, it induced intestinal dysfunction, such as increased intestinal permeability in the absence of mucosal damage. Importantly, the authors found that the intestinal inflammation could be reduced by daily gavage with Lactobacillus, which were able to degrade ATIs. As reported above, our group has shown that ATIs from *T. monococcum* are degraded during digestion by gastric-duodenal enzymes, compared to ATIs from common wheat. Therefore, assuming that dietary ATIs may also modulate intestinal permeability in patients with wheat-related disorders, we can hypothesize that a regular diet based on *T. monococcum* in such patients, might prevent gut barrier damage.

## 5. Beyond Immunogenicity: Nutritional Features of Diploid Wheats

Increasing attention to the nutritional aspects of food has led to the search for alternatives to the traditional *T. aestivum* wheat. The HEALTHGRAIN project showed a different composition of dietary fiber, polyphenols, minerals, trace elements, vitamins, carotenoids, and alkylresorcinols by comparing ancient and modern wheats [65].

Compared to modern wheats, diploid wheats showed a better nutritional quality and relevant potential for human consumption. In particular, ancient wheats contains higher levels of antioxidant compounds as α- and β-carotenes, lutein, zeaxanthin, tocols, conjugated polyphenols, alkyl resorcinols and phytosterols, retinol, phosphorus, potassium, riboflavin, and pyridoxine [66,67,68,69].

Moreover, ancient wheats had a lower quantity of dietary fibre and carbohydrate, but a higher content of proteins, lipids (mostly unsaturated fatty acids), fructans, thiamine and a number of other B vitamins, zinc, and iron [66,67,68,69,70], compared to modern wheats, which gives them properties useful in preventing some pathological conditions.

Although concrete functional benefits are difficult to ascertain, some results from human trials suggested that, compared to the consumption of products made from modern varieties, the consumption of products made with ancient wheat varieties ameliorate pro-inflammatory/anti-oxidant parameters, as well as glycaemic and lipid status [71]. 

Ancient wheats are still cultivated today, but only in some areas of the world, including France, Germany, Austria, Hungary, Bulgaria, and Italy; however, the increasing interest to healthier foods has increased the popularity of their use and, consequently, has caused an increase in their production.

Beyond their improved health benefits, despite the lower grain yields of these ancient wheats, several studies have underscored a good bread-making quality and a good usefulness for the preparation of cookies and good-quality pasta [72].

For all these features, the use of ancient wheats may become more relevant in human consumption, especially in the development of new or special functional foods, with superior nutritional quality.

## 6. Conclusions

Gluten, and other wheat proteins including ATIs and FODMAPs, have been identified as possible factors for the generation of intestinal and extra-intestinal symptoms in subjects suffering from wheat-related disorders, such as NCWS. In particular, it is well-known that gluten and ATIs possess immune stimulating activity. Therefore, dietary exposure to the combination of gluten and ATIs exacerbates intestinal immune dysregulation, and increase risk to develop wheat-related disorders. *T. monococcum*, the oldest and most primitive cultivated wheat, unexposed to genetic improvements, has been suggested to possibly exert a reduced immunostimulating activity compared to common wheats and, consequently, embodies the role of a fitting candidate to be introduced into the diet of such patients. Therefore, clinical studies on NCWS patients, to assess the effects of a TM wheat–based food diet, are warranted.

## Figures and Tables

**Figure 1 cells-11-02389-f001:**
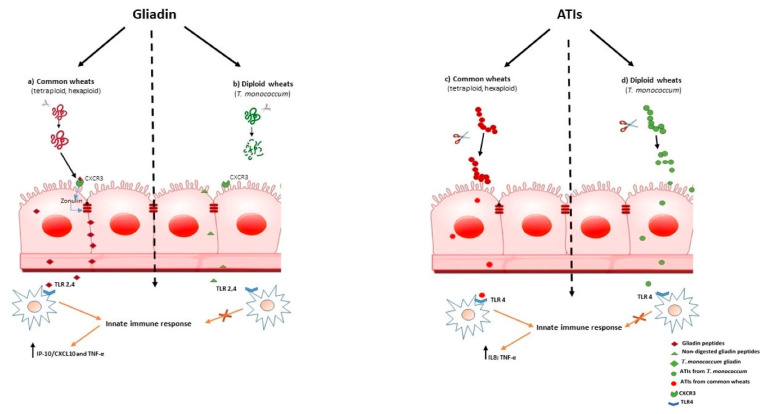
Schematic drawing that shows what happens in subjects with non-coeliac wheat sensitivity on *T. monococcum* based diet containing common wheat, according to our hypothesis. (**a**) Interactions between non-digested gliadin peptides from common wheat and CXCR3 receptors in the intestinal epithelium trigger zonulin, release that leads to increased intestinal permeability. Therefore, non-digested gliadin peptides can reach the lamina propria and could activate the innate immune system via TLR-2 and TLR-4 receptors, inducing the release of pro-inflammatory cytokines; (**b**) As *T. monococcum* gliadin have a marked susceptibility to gastro-intestinal digestion, it can be hypothesized that such a mechanism, triggered by non-digested gliadin from common wheats, may not be elicited. Amylase trypsin inhibitors (ATIs) have been shown to be potent activators of the innate immune system in NCWS subjects. (**c**) ATIs from common wheats pass the intestinal epithelium and in LP stimulate TLR4 on macrophages, inducing the production of innate cytokines; (**d**) Considering that *T. monococcum* contains ATIs with a higher digestibility than modern wheat, the innate immune response could be prevented. Therefore, TM could retain a lower immunostimulating activity for subjects suffering from NCWS.

**Table 1 cells-11-02389-t001:** Gut immune activation in NCWS.

Gut Immune Activation in NCWS	References
**Innate immune response** expression of TLR2 production of innate immune cytokines	Sapone et al., 2011 [7]; Sapone et al., 2011 [7]; Lammers et al., 2011 [8]; Junker et al., 2012 [9]; Vazquez-Roque et al., 2013 [10]; Di Liberto et al., 2016 [11]; Caminero et al., 2016 [12]; Zevallos et al., 2017 [13]; Iacomino et al., 2021 [14]; Cárdenas-Torres et al., 2021 [15];
**Adaptive** **immune response** production of IFN-γ production of TNF-α and IL-17	Brottveit et al., 2013 [17] Mansueto et al., 2020 [18]; Castillo-Rodal et al., 2020 [19]
**Autoantibodies** production of antigliadin IgG antibodies	Sapone et al., 2011 [7]; Carroccio et al., 2012 [20]; Volta et al., 2012 [21]; Uhde et al., 2016 [22]
**Intestinal permeability** high levels of CLDN4 Increased transepithelial electrical resistance break of tight junctions and infiltration of the intestinal epithelium by T cells high levels of multiple permeability biomarkers (LBP, FABP2)	Sapone et al., 2011 [7] Hollon et al., 2015 [23] Fritscher-Ravens et al., 2014 [24] Uhde et al., 2016 [22]
**Mucosal immune cells** increased infiltration of eosinophils increased levels of mast cells intraepithelial lymphocytosis	Carroccio et al., 2019 [20]; Zanini et al., 2018 [25] Losurdo et al., 2017 [26]; Giancola et al., 2020 [27]; Sapone et al., 2010 [7]; Brottveit et al., 2013 [17]; Volta et al.,2012 [21]; Carroccio et al., 2012 [20]; Carroccio et al., 2019 [28]; Zanini et al., 2018 [25]; Rostami et al., 2022 [29]

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
