# Peer review of "Diploid Wheats: Are They Less Immunogenic for Non-Celiac Wheat Sensitive Consumers?"

_cells, 2022, doi:10.3390/cells11152389_

Round 1

Reviewer 1 Report

The topic is nice but I have few questions concerning the modern wheat did  the authors mean that the wheat is genetically modified and is it the same for diploid, triploid wheat? 

T. monococcum is the brown wheat grain ?

Table 1 is not very clear, authors should modify it.

Author Response

Reviewer #1:

  1. The topic is nice but I have few questions concerning the modern wheat did the authors mean that the wheat is genetically modified and is it the same for diploid, triploid wheat? 

Authors

For modern wheats we mean the polyploid Triticum wheats consisting of tetraploid and hexaploid species, not genetically modified

  1. monococcum is the brown wheat grain ?

Authors

Yes

  1. Table 1 is not very clear, authors should modify it.

Authors

Done

Reviewer 2 Report

Comments:

Review of the paper

cells-1806514

This paper aimed to review the toxicity of diploid wheat immunogenicity to non-celiac wheat sensitive consumers. The manuscript is well written, and the results are well discussed. However, several issues should be solved before resubmitting a revision for being reviewed again.

GENERAL ISSUE

- Previous studies have summarized that wheat related proteins may be the main cause of non-celiac disease, and amylase trypsin inhibitor, as a protein in wheat endosperm and flour, can also cause intestinal related symptoms. A large number of studies on the immunogenicity of wheat related proteins were summarized in this review, and the immunogenicity of wheat related proteins and related symptoms were classified and summarized, but the characteristics and immunogenicity of diploid wheat were not well highlighted. Since the authors emphasized the Immunogenic properties of diploid wheats in the title, they should compare the characteristics and immunogenicity of diploid wheats with commom wheats throughout the manuscript, including the Table and Figures.

TABLE

- Table 1: This table was not clear for viewing. Authors should prepare a figure (use some symbols, not use words only) instead of table to show the gut immune activation in NCWS.

- Table 2: The authors should add more details (such as the data in the literature) when comparing the features of monococcum wheats with common wheats.

FIGURE

- Figure 1: There were two “Figure 1” in the title. I cannot even see the texts in the figure. This figure was also not clear for viewing. Authors should prepare a figure with high quality and resolution.

Author Response

Reviewer #2:

  1. Since the authors emphasized the Immunogenic properties of diploid wheats in the title, they should compare the characteristics and immunogenicity of diploid wheats with commom wheats throughout the manuscript, including the Table and Figures.

Authors

As suggested by the Reviewer we have emended the text comparing the characteristics and immunogenicity of diploid wheats with commom wheats throughout the manuscript, including the Table and Figures. (manuscript marked; page 5, line 189; page 6, lines 212 and 237; page 8, lines 300; page 10, lines 357-358)

  1. Table 1: This table was not clear for viewing. Authors should prepare a figure (use some symbols, not use words only) instead of table to show the gut immune activation in NCWS.

Authors

We have increased the resolution of Table 1.  Moreover, the text contain Figure 1 that show the gut immune activation in NCWS.

  1. Table 2: The authors should add more details (such as the data in the literature) when comparing the features of monococcum wheats with common wheats.

Authors

We are grateful to the Reviewer for the suggestion therefore in line with his/her suggestion, we have emended the text presenting more details in Table 2 according to the Reviewer’s input.   

  1. Figure 1: There were two “Figure 1” in the title. I cannot even see the texts in the figure. This figure was also not clear for viewing. Authors should prepare a figure with high quality and resolution.

Authors

We have corrected the mistake in the title,  included the text and increased the resolution of Figure 1, as suggested by the reviewer.

Reviewer 3 Report

Very meaningful discussion on a rare/difficult disease, with great language and logic.

For the title:

Two points for the authors to consider

non-celiac wheat “sensitivity” -> sensitive?

Is it really proper to call this toxicity?

toxic -> Immunogenic? Or immune stimulating activity?

Table 1 needs “Reference” as the title of the 2nd column

Table 1 needs proper in-text reference (etc 17, 18, 19… 29, 30, 31) in column 1

Same for table 2

L185 extra space at the very beginning

L289 extra text of “Figure 1”

Figure 1 needs a higher resolution picture

L 358: subjects-> patients?

Author Contributions session not completed.

Author Response

Reviewer #3:

  1. For the title:

Two points for the authors to consider

non-celiac wheat “sensitivity” -> sensitive?

  1. Is it really proper to call this toxicity?

toxic -> Immunogenic? Or immune stimulating activity?

Authors

We are grateful to the Reviewer for the suggestions therefore we have modified the title and text in according to the Reviewer’s input.  (manuscript marked page 1, line 13; page 7, lines 269-270; page 8, lines 301-302; page 10, line 357)

  1. Reviewer #3:

Table 1 needs “Reference” as the title of the 2nd column

Authors

Done

  1. Table 1 needs proper in-text reference (etc 17, 18, 19… 29, 30, 31) in column 1

Same for table 2

Authors

Done

  1. L185 extra space at the very beginning

Authors

Corrected

  1. L289 extra text of “Figure 1”

Authors

Corrected

  1. Figure 1 needs a higher resolution picture

Authors

Done

  1. L 358: subjects-> patients?

Authors

We have emended the text reporting “patients” in place of “subjects” (manuscript marked page 10, line 359)

  1. Contributions session not completed.

Authors

We are grateful to this Reviewer for having underlined this inattention, therefore, the Author Contributions session has been completed.

Round 2

Reviewer 2 Report

The authors have revised the manuscript accordingly, except the comment "3. Table 2: The authors should add more details (such as the data in the literature) when comparing the features of monococcum wheats with common wheats.". I have not seen the data in the literature in this table. If authors just described the features of monococcum wheats in Table 2 only with words (no data were shown), there is no need to prepare this table, describing these features in the main text is better.  

Author Response

As suggested by the Reviewer’s we have deleted the table 2, describing the features of monococcum wheats in the main text (page 9, lines 315-348)